# Medics as a Positive Deviant in Influenza Vaccination: The Role of Vaccine Beliefs, Self-Efficacy and Contextual Variables

**DOI:** 10.3390/vaccines10050723

**Published:** 2022-05-05

**Authors:** Dorota Włodarczyk, Urszula Ziętalewicz

**Affiliations:** Department of Health Psychology, Medical University of Warsaw, 00-575 Warsaw, Poland; dorota.wlodarczyk@wum.edu.pl

**Keywords:** psychosocial factors, influenza vaccination, medics, non-medics

## Abstract

The influenza vaccination rate remains unsatisfactorily low, especially in the healthy adult population. The positive deviant approach was used to identify key psychosocial factors explaining the intention of influenza vaccination in medics and compare them with those in non-medics. Methods: There were 709 participants, as follows: 301 medics and 408 non-medics. We conducted a cross-sectional study in which a multi-module self-administered questionnaire examining vaccination beliefs, risk perception, outcome expectations (gains or losses), facilitators’ relevance, vaccination self-efficacy and vaccination intention was adopted. We also gathered information on access to vaccination, the strength of the vaccination habit and sociodemographic variables. Results: We used SEM and were able to explain 78% of the variance in intention in medics and 56% in non-medics. We identified both direct and indirect effects between the studied variables. In both groups, the intention was related to vaccination self-efficacy, stronger habits and previous season vaccination, but access to vaccines was significant only in non-medics. Conclusions: Applying the positive deviance approach and considering medics as positive deviants in vaccination performance extended the perspective on what factors to focus on in the non-medical population. Vaccination promotion shortly before the flu season should target non- or low-intenders and also intenders by the delivery of balanced information affecting key vaccination cognitions. General pro-vaccine beliefs, which may act as implicit attitudes, should be created in advance to build proper grounds for specific outcome expectations and facilitators’ recognition. It should not be limited only to risk perception. Some level of evidence-based critical beliefs about vaccination can be beneficial.

## 1. Introduction

According to the World Health Organization [1], the influenza vaccine is an effective prevention against seasonal influenza and is recommended for healthy adults and specific risk groups. Despite the availability of safe vaccines and influenza’s extensive outcomes, vaccine uptake in the general public and high-risk groups is rather low. In Europe [2], the median for healthcare workers was 30.2%. For example, in Poland, 4.1% of the population was vaccinated against influenza in the season 2019/2020 [3], which was one of the lowest flu vaccination rates, similar to previously reported seasons [2].

To determine factors that increase influenza vaccine uptake in the general public, we used positive deviance, an asset-based approach that learns from those who demonstrate better performance in an outcome of interest by identifying their strengths and resources and exploring how and why things go right [4]. Although the implementation of influenza vaccinations among medics is not satisfactory, it is higher than in the general public (non-medics), and this group can potentially be treated as a positive deviant.

The two groups obviously differ in the type of education, but both consist of healthy adults who not only are at risk of infection and its outcomes but also play an important role in virus transmission [1]. It was confirmed that medics are not free from vaccination hesitancy [5,6], and, paradoxically, being the most trusted information sources by the public, some of them are losing confidence in vaccines [7].

A recent systematic review showed differences in vaccination barriers between the risk groups and little contribution from most sociodemographic variables [8]. This confirmed the predictive power of selected psychosocial variables, which allowed effective, evidence-informed interventions to be designed. Studies also identified factors specific to medics [9].

The most influential theories used in studies determining vaccination intention and performance are the health belief model [10], the theory of planned behaviour [11], and social cognitive theories [12] or the Health Action Process Approach (HAPA; [13]). Although there is considerable overlap between them and they differ slightly in terminology, they indicate key relevant constructs and assume direct and indirect relationships between them [14]. In stage models, it is expected that different factors play a role, depending on the stage of change [15].

In line with these theories, many studies confirmed that risk perception predicted vaccination uptake [16]. Usually, it addresses the relative vulnerability and compares the risk of having the flu or negative consequences of the disease compared to people of the same age and gender [2,17]. Perceiving a low risk of the disease was identified as a barrier to influenza vaccine uptake [8], suggesting that this factor is rather stage-specific and plays a role when the motivation to change develops but is insufficient to enable a person to change behaviour [18].

A number of studies referred to attitudes and beliefs towards vaccinations, showing that misconceptions, hesitancy or an antivaccination approach were associated with poor performance [19]. This can be considered at the following two levels: general beliefs (positive or negative) include convictions about influenza, understanding of the safety and effectiveness of vaccines, unique theories about the purposes of vaccination, civil obligations and liberties or trust in scientific authorities [20]; specific outcome expectancies include individuals’ perception of links between action and subsequent outcomes and the specific gains and losses resulting from vaccination [18].

Another factor proved to significantly impact influenza vaccination intention [21] and implementation [22] was self-efficacy, which refers to individuals’ beliefs in their capabilities to exercise control over new behaviour and their own functioning even in the face of barriers [18].

It was shown that social and environmental facilitators, such as encouragement by supervisors or well-organized vaccination campaigns with on-site vaccination [23] and receiving reminders to vaccinate (e.g., a text message), had statistically relevant effects on increasing influenza vaccination rates [24].

Some contextual factors also played a role. The previous seasonal influenza vaccination was found to be a positive predictor of vaccine uptake [25,26]. Many research results, also in medics, confirmed the positive effects of access to vaccination but this can be meaningless when individuals have to pay for it, especially in poorer countries [27].

Drawing from the theories of health behaviour and the existing results on influenza vaccination determinants, we selected the key factors related to the vaccination intention in order to compare them between medics and non-medics. Following HAPA [28], we focused on the paths connecting self-efficacy with a behavioural intention on the one hand and risk perception and various beliefs about vaccines on the other hand. The conceptual model tested in the study is presented in Figure 1. We assumed that the different interrelated types of vaccine beliefs may influence self-efficacy and can give way to the intention [18,29]. In line with the attitude concept [30,31], we focused on more general and more specific beliefs about influenza vaccination, which constituted ‘distal’ and ‘proximal’ antecedents of self-efficacy. The general pro- and antivaccine beliefs refer to convictions about flu vaccination in general without personal commitment. The influenza vaccination-specific beliefs assume personal commitment and refer to perceived gains and/or losses resulting from vaccination (also labelled as outcome expectations) and perceived facilitators and/or barriers.

The aim of the study was to test the model in medics and non-medics and compare the effects between the groups. We hypothesized the following: (1) general pro- and antivaccine beliefs together with risk perception would be related to self-efficacy through influenza vaccination-specific beliefs; (2) self-efficacy would mediate the relationships between vaccine beliefs and behavioural intention; (3) behavioural intention would be related also to previous flu vaccination experiences and access to vaccines; (4) the relationships between variables in the model will differ between medics and non-medics.

## 2. Materials and Methods

### 2.1. Participants

The results were obtained from 709 participants forming two groups: medics (*n* = 301) and non-medics (*n* = 408). They were recruited from the voluntary registered panel participants who gave informed consent before participating in a study and met the criteria of inclusion. Medics were significantly older than non-medics (M = 52.10, SD = 12.43 vs. M = 41.04, SD = 13.64; *t*_(676, 24)_ = 11.23, *p* < 0.001). The number of female participants was higher among medics (66.1% vs. 50.7%; χ^2^_(1, *n* = 709)_ = 16.74, *p* < 0.001). There was also a significant difference in the distribution of education (χ^2^_(2, *n* = 709)_ = 212.00, *p* < 0.001). All medics had higher education at the level of at least Master’s degree (82.7.1%) or Bachelor’s degree (17.3%). Among non-medics, the majority had secondary education (40.4%), followed by primary or vocational education (29.7%) and higher education (29.9%). The groups differed also in size of living location (χ^2^_(4, *n* = 709)_ = 21.62, *p* < 0.001). Among non-medics, inhabitants of rural areas (34.1%) and medium-sized towns (28.7%) were more numerous than inhabitants of other areas. Among medics, the distribution was quite even. Medics performed the following professions: doctor (*n* = 151, 50.2%), nurse (*n* = 106, 35.2%), physiotherapist (*n* = 4, 1.3%), midwife (*n* = 7, 2.3%) and paramedic (*n* = 33, 11.0%).

### 2.2. Measures

The study instrument was a multi-module self-administered questionnaire constructed according to recommendations in the field [32,33]. The questions were built according to an algorithm; these are strictly structured sentences in which the part concerning a given type of behaviour can be changed (the translated version of the questionnaire is available in the Appendix A). Content validity was established by sending the questionnaire to a panel of five experts. Questions with a mean standard deviation of experts’ opinions greater than 1.1 were removed. Reliability was confirmed using Cronbach’s alpha and confirmatory factor analysis (CFA) for each module [34].

The module measuring general vaccination beliefs has two subscales: pro-vaccine beliefs (VacPros) comprising three items on vaccine security, trust in science and pro-social attitude (sample item: It is safer to be vaccinated against the flu than to get it); antivaccine beliefs (VacCons) comprising three items on undesirable effects, low seriousness/prevalence of influenza and dishonesty of the vaccine industry (sample item: The harmfulness of the side effects of influenza vaccinations is greater than the resulting benefits). The responses were rated on a five-point scale (1 = strongly disagree; 5 = strongly agree). Reliability of the VacPros subscale is α = 0.80 for medics and α = 0.89 for non-medics; for the VacCons subscale, reliability is α = 0.76 for medics and α = 0.81 for non-medics. Indices of model fit in CFA were the following: RMSEA = 0.044, SRMS = 0.019 and CFI = 0.994.

The module measuring risk perception (Risk) refers to relative risk of having flu or its consequences. It contains four items in which participants rate their individual risk compared to others on a five-point scale from 1 (definitely lower risk than others) to 5 (definitely higher risk than others). Sample item: Compared to others of the same sex and age as you, how would you estimate the likelihood that in your current situation you will have the flu. Reliability for medics is α = 0.89 and for non-medics is α = 0.89. CFA indices are the following: RMSEA = 0.076, SRMR = 0.007 and CFI = 0.983.

The module measuring outcome expectations encompasses perceived gains (VacGains) and perceived losses (VacLosses). The VacGains subscale consists of eight items on perceived gains from influenza vaccination with a five-point rating scale from 1 (strongly disagree) to 5 (strongly agree). Sample item: Influenza vaccination reduces my risk of getting the virus and/or getting the flu. Reliability is α = 0.93 for medics and α = 0.94 for non-medics. CFA indices are the following: RMSEA = 0.079, SRMR = 0.023 and CFI = 0.982. The VacLosses subscale consists of nine items measuring perceived losses from influenza vaccination with a five-point rating scale (1 = strongly disagree; 5 = strongly agree). Sample item: My relatives would criticize me or be dissatisfied with me. Reliability for medics is α = 0.80 and for non-medics is α = 0.88. CFA indices are the following: RMSEA = 0.072, SRMR = 0.038 and CFI = 0.961.

The module measuring facilitators’ relevance (FacRel) includes factors potentially conducive to influenza vaccination. The scale consists of nine items on the following question: Regardless of whether you vaccinate or not, does or could this factor favour your decision to vaccinate? Sample item: Publicly available information to remind you when and how to vaccinate. The responses are rated on a five-point scale (1 = definitely irrelevant to me; 5 = definitely relevant to me). Reliability for medics is α = 0.89 and for non-medics is α = 0.94. CFA indices are the following: RMSEA = 0.071, SRMR = 0.026 and CFI = 0.977.

The vaccination self-efficacy scale (VacSE) consists of six items on a person’s belief in the extent to which they are able to perform vaccination, even though difficulties arise. A five-point scale was used (1 = definitely not sure; 5 = definitely sure). Sample item: To what extent are you sure that you will vaccinate in the current season even if you have to pay in full or in part for the influenza vaccination? Reliability for medics is α = 0.98 and for non-medics is α = 0.97. CFA indices are the following: RMSEA = 0.078, SRMR = 0.007 and CFI = 0.994.

The vaccination intention (Intention) was established by asking if a person had an intention to vaccinate in the current season, with answers selected from the following: I have already vaccinated (vaccination implementation) to I am not going to get vaccinated. Additionally, data on the contextual variables (last year’s influenza vaccination, five-year vaccination, vaccine access) and age, gender and education were collected.

### 2.3. Procedure

A cross-sectional study was conducted between October and December 2020 (during the second wave of the COVID-19 pandemic in Poland) by a professional survey company. The company secured selection of the participants from those who freely registered with the panel and gave necessary (true and up-to-date) information to participate in studies run by the company together with their informed consent. Non-medics filled in the above multi-module survey online. Medics participated in the computer-assisted telephone interview (CATI), chosen as the method for securing the highest participation rate at that time (lockdown with low number of patients in facilities and not frequently used teleconsultations joint with low computer proficiency in medics). The inclusion criteria for medics were the following: being a professionally active family doctor or pediatrician (50% of the group); or a nurse, midwife, paramedic or physiotherapist; for non-medics, meeting the age criteria representative of the general public in the country.

### 2.4. Statistical Analysis

The statistical analysis was conducted using IBM SPSS Statistics 26 and Mplus for the SEM. The a priori sample size for structural equation modelling (SEM) was calculated based on the following assumptions [35]: anticipated effect size = 0.3, statistical power level = 0.8, number of latent variables = 45, number of observed variables = 13, probability level = 0.05 [36,37]. It was also based on the sample-to-variable ratio, which suggests an observation-to-variable ratio of 15:1 or 20:1 [38]. Regardless of the method of estimation, the achieved sample size in each group was sufficient.

The first step of the analysis was establishing the equivalence of scales between medics and non-medics by measuring both metric and scalar invariance [39]. The procedure is highly recommended when different social groups or groups tested under different conditions are compared (e.g., interview versus online). The criteria of invariance were as follows [40]: change in CFI ≤−0.010 paired with RMSEA ≤0.015 or SRMR ≤0.030 (metric invariance); change in CFI ≤−0.010 paired with RMSEA ≤0.015 or SRMR ≤0.010 (scalar invariance). All scales proved to be equivalent and proper for further comparisons between medics and non-medics.

Descriptive statistics were used to characterize the sample. Chi-square tests were performed to examine group differences in frequencies. Student’s *t*-test or the Mann–Whitney U test was used to compare normally or non-normally distributed variables, respectively. To examine correlations between variables we used Pearson’s coefficient. In order to verify differences in relationships between predictors of vaccination intention in medics and non-medics, we used multi-group SEM [41,42]. The results for both groups are presented in Figure 2, with ovals and rectangles representing latent and observable variables, respectively, and values of path coefficients located above (for non-medics) and below (for medics) the arrows representing the strength of the relationships between variables.

## 3. Results

### 3.1. Differences between Groups in Studied Variables

The frequencies of responses reflecting readiness for influenza vaccination are presented in Figure 3. Medics had higher readiness for getting vaccinated than non-medics (Me = 1.00 and Me = 5.00, respectively, U = 29928.00 and *p* < 0.001 after excluding subjects with contraindications).

At the end of December 2020, 48% of medics were vaccinated (versus 8% of non-medics) and the same percentage of non-medics declared they were not going to get vaccinated (versus 24% of medics).

The groups differed significantly in the levels of predictors included in the study, with the exception of Risk (Table 1). Medics were higher in VacPros, VacGains, FacRel and VacSE and lower in VacCons and VacLosses. Risk below three indicates a slight bias towards defensive optimism in both groups.

A total of 76% of medics and 30% of non-medics declared getting vaccinated in the previous influenza season (χ2_(1)_ = 81.14, *p* < 0.001). However, there were missing data in the answers to the following question: *n* = 89 (30%) in medics and *n* = 235 (58%) in non-medics. Medics declared more frequent vaccination in the last 5 years than non-medics (U = 89,888.5, *p* < 0.001; Me = 4 and Me = 1, respectively) and better access to influenza vaccinations (U = 19,750.00, *p* < 0.001; Me = 1.00 and Me = 4.00, respectively; reversed scale).

### 3.2. Verification of the Model

Using SEM allowed us to determine whether there are significant differences between the groups by comparing the fit indices between the models with certain parameters constrained to be equal and a model with those same parameters freely estimated (allowed to differ) across the groups. In the first step, we aimed to confirm that assuming there were no differences between medics and non-medics would be wrong. To this end, we tested Model 1 with all parameters allowed and Model 2 with all parameters constrained. The models differed significantly (*p* < 0.001). Model 1 (fully unconstrained) obtained a significantly better fit (χ^2^_(2216)_ = 4291.946, RMSEA = 0.051, CFI = 0.915, SRMR = 0.070) than the fully constrained Model 2 (χ^2^_(2266)_ = 4536.115, RMSEA = 0.053, CFI = 0.907, SRMR = 0.112). These results indicate that the relationships between the determinants of vaccination intentions are different for medics and non-medics. Thus, the next step in our analysis was explorative.

### 3.3. Comparison of the Model in Medics and Non-Medics

In order to compare the fit indices of the entire theoretical model and to explore the nature of the differences in the model between medics and non-medics, we analysed the model for each group separately. Our model fitted the data well in both groups, as summarized in Table 2.

### 3.4. Direct Predictors of Vaccination Intention/Implementation and Self-Efficacy

A table showing the correlation coefficients between the studied variables can be found in Appendix A. The results of the SEM conducted separately for each group are presented in one figure to enable tracking similarities and differences between groups. As shown in Figure 2, the model explains a noticeable percentage of the Intention variance as follows: 78% of medics and 56% of non-medics. The medics’ model included a lower number of significant associations, but they were stronger than in the non-medics’ model. In neither group was age, gender or education a predictor of Intention.

In both groups, the Intention was related to VacSE, and the strength of these connections was similar (0.45 and 0.48 for non-medics and medics, respectively). The stronger habit (5-year vaccination) and previous season vaccination were also predictors of Intention, but low access to vaccines was significant only in non-medics (path coefficient −0.18).

The remaining factors in the model explained the variance of VacSE better in medics than in non-medics (64% and 49%, respectively). In both groups, the direct predictors of VacSE were VacGains (stronger in medics) and FacRel. However, only in non-medics was the role of VacLosses significant (path coefficient −0.54). The high VacLosses are related to low VacSE.

VacPros directly and positively predicted VacGains and FacRel in both groups, but more strongly in medics (1.69 versus 0.66 in non-medics). The roles of VacCons and Risk differed substantially between the groups. In medics, VacCons directly and positively predicted both FacRel and VacGains (1.63 and 0.77, respectively) and were not related to VacLosses. In non-medics, they directly and positively predicted VacLosses (path coefficient 0.53) and were negatively connected with FacRel and VacGains (−0.19 and −0.12, respectively). Only in non-medics did Risk significantly predict FacRel and VacGains (the association with VacLosses approached significance).

### 3.5. Indirect Effects

Table 3 shows that the groups differed in the number of indirect effects confirmed in the model. The path from FacRel through VacSE to Intention was significant in both groups. The same refers to the effects of VacPros on VacSE with VacGains and FacRel as mediators; however, they were stronger in medics. The paths from VacCons and Risk (through VacLosses and FacRel) to VacSE were specific only to non-medics. The effect of VacCons on VacSE through VacGains was significant in medics and only approached significance in non-medics. However, their directions were the opposite. In medics, VacCons were positively related to VacGains, which in turn predicted VacSE. In non-medics, the association between VacCons and VacGains was negative and rather weak.

## 4. Discussion

The data collected at the beginning of the flu season during the COVID-19 pandemic allowed us to capture not only the flu vaccination intention but also vaccination implementation in part of the sample. The results confirmed that medics performed much better than non-medics and that proportions of readiness to hesitancy were almost inverse in the groups. This finding confirmed our supposition that the group of medics may be considered a positive deviant in vaccination performers. By design, the groups differed in education, but the medics were also older, with a slight predominance of females and those living in bigger towns. We intentionally focused on medics who, due to their unique education and experience, are expected to behave in line with medical knowledge. The problem is that medical education is not a sufficient condition for influenza vaccination, and there are other factors which may facilitate or hinder this behaviour. Believing that medical education may promote some process that increase vaccination intention, we wanted to analyse this process and see if it would be possible to activate it among people not trained professionally in medicine. Thus, we treated the potential bias as an opportunity.

This readiness rate in medics, which is rather high in relation to previous seasons, can be attributed to the COVID-19-related increase in vaccination motivation [43]. However, the effect confirmed in the general population of the United Kingdom [44] was only observed to a small extent in participating non-medics, which may suggest that medics are more responsive to external cues for adjusting their health behaviour. Even so, approximately 30% of medics declared vaccination refusal or hesitancy, which clearly shows that the higher vaccination uptake in medics cannot be attributed solely to their education.

As for the first hypothesis, general pro- and antivaccine beliefs together with risk perception were related to self-efficacy through influenza vaccination-specific beliefs, but there were differences between groups. The variance of vaccination self-efficacy in medics was fully explained only by the paths positively connecting pro- and antivaccine beliefs with both gains and facilitators.

In non-medics, antivaccine beliefs were related positively to perceived losses and negatively to gains and facilitators, whereas in medics the first effect did not exist, and the last ones were positive. One can speculate that these positive relationships in medics may be the result of unbiased knowledge of vaccines, reasons and likelihood of adverse events, procedures of testing vaccines and, finally, realistic expectations and evidence-based critical thinking. This would suggest that there are some benefits to the doubts [45] and that some levels of negative beliefs about vaccines can be conducive to perceiving gains and facilitators. The other option would be that it is not a matter of the level of negative beliefs only, but rather the proportion between positive and negative beliefs, similar to the positivity ratio [46,47]. For example, the pro-/antivaccine belief ratio in medics was 2.65, whereas in non-medics it was 1.22. Further investigation is needed to explore if this indicator would be applicable to general vaccination beliefs. The gains/losses ratio can also be considered [48]. Generally, it is believed that gain-framed messages can be more effective when advocating prevention behaviour such as vaccination [49]. On the other hand, some groups of people for whom loss-framed messages can be more successful, depending on their level of involvement or knowledge, should not be ignored [50]. Probably a more balanced message emphasising gains but not omitting losses could be effective.

In non-medics, antivaccine beliefs are also negatively related to facilitators. This connection was even stronger than in medics, which suggests that vaccination campaigns and interventions for the general public should focus on facilitators such as personalised reminders, positive modelling by a doctor, a conversation initiated by a doctor, encouragement by supervisors or free access and friendly procedures [23].

Although the groups did not differ in their levels of risk perception, to medics they were irrelevant. According to HAPA [18], risk perception is the determinant of the intention specific to the motivational stage. This seems to be advantageous mainly initially to put people on track to developing the motivation to change, but later on, other variables are more influential in the self-regulation process. In non-medics, risk perception played a positive role. The above observations would suggest that medics were at a more advanced stage of behavioural change, where some factors, such as risk perception and vaccination losses, no longer played a significant role. According to the precaution adoption process model [51], it seems that non-medics predominantly might be still in one of four following stages: unengaged; undecided; decided not to act; or decided to act. Medics are likely to be in one of the following stages: decided to act; acting (need resources to act, detailed ‘how-to’ information and self-efficacy); or maintenance. If yes, then the question arises about factors promoting the transition from one stage to another. Assuming that people at different stages represent a stage-specific mindset and need to master different tasks, it seems that intervention tailored to the individual’s stage would be the most appropriate [15]. On the other hand, it cannot be ruled out that some stages differ in the levels of some predictors only quantitatively (e.g., each successive stage shows significantly higher self-efficacy or a stronger intention to perform vaccination).

As for the second hypothesis, it was confirmed in both groups. Self-efficacy mediated the relationships between vaccine beliefs and behavioural intention. In medics, the main mechanisms regulating vaccination intention/implementation seem to rely on supporting perceptions of vaccination gains and facilitators’ relevance. The positive connections existed also in non-medics but were weaker. This indicates that all these vaccine cognitions should be enhanced because they capture the most relevant factors in the 4C (confidence, convenience, complacency, calculation) model of vaccine hesitancy [52].

Further differences between groups were observed in direct predictors of vaccination self-efficacy. Especially intriguing was the path running through perceived losses. The very tempting supposition would be that as long as the levels of losses were as high as in non-medics, they were potent enough to decrease self-efficacy, mediating the relationships with antivaccine beliefs. These beliefs, including undesirable vaccine effects, low seriousness/prevalence of influenza and dishonesty of the vaccine industry [53,54], were connected to potentially negative outcomes expected after vaccination, such as social rejection, everyday inconveniences and health risks. The problems of uncertainty about the safety of flu vaccines and mistrust of information provided to the public that dismisses concerns were previously noticed [45].

The third hypothesis refers to the contextual variables. Apart from vaccine access, the relationships between vaccination intention/implementation and the most proximal predictors were similar in both groups. These results are in line with other studies confirming the positive effects of vaccination self-efficacy [21,22,55,56], previous experiences (or habit) and last season’s vaccination [8,25,57,58,59] both in medics and other groups. In medics, vaccine access turned out to be insignificant, probably due to low variance in this variable; as a priority group, they had easier access than the general public. Additionally, having to pay for vaccines is a proven barrier to vaccination [27]. In none of the groups were the included sociodemographic factors significant contributors.

In the light of the above discussion, we may confirm the fourth hypothesis regarding the differences in the relationships tested in the model between the groups.

Some limitations of the study should be mentioned. Vaccination performance was not the only difference between the groups, and there might be further contributors to vaccination intention that should be included in the model. Importantly, the groups also differed in some sociodemographic characteristics (medics were older, with a slight predominance of females and those living in bigger towns). These factors may influence general and specific beliefs about vaccination and distort differences between the study groups. Importantly, the analysis confirmed the equivalence of scales between the groups, and these factors were not related to the variables in the model when testing each group separately. However, the conclusions should be treated with caution.

The numbers in different medical groups were unequal. The self-reported data can be biassed by social approval, especially as vaccination has recently become a controversial topic [60]. The study design was cross-sectional, which does not allow the hypothesised causal relationships between predictors to be confirmed. The results, therefore, should be treated as explorative. The number of potential contributors to vaccination intention is much higher than those included in our study. We started with a truncated model; for example, the intention–behaviour gap was not analyzed. Among those who form an intention, more than half fail to put this intention into practice. This should be analysed in further studies using the positive deviant approach.

## 5. Conclusions

It seems that the positive deviant approach adopted in this study allowed us to broaden our perspective and be better informed on the specificity of the better performer of influenza vaccination. It was probably the first attempt to use it in the vaccination context. It seems that providing comprehensive and up-to-date information about the risks and benefits of vaccination to the public is essential. Reliable knowledge of vaccines together with clearing up controversies and misconceptions should be complemented by trust-building strategies with information on gains (gains should noticeably outweigh losses) and facilitators (‘how-to’ and ‘just do’). Among those with high pro-vaccine beliefs, some level of evidence-based critical beliefs about vaccination can also be beneficial.

In the case of influenza vaccination, which is implemented annually, the individual’s focus on performance varies over time. Vaccination promotion shortly before the flu season should target non- or low-intenders and also intenders by the delivery of a balanced scope of information affecting key vaccination cognitions. Additionally, general pro-vaccine beliefs, which may act as implicit attitudes, should be created in advance to allow priming processes after delivering information on flu vaccine availability [61].

## Figures and Tables

**Figure 1 vaccines-10-00723-f001:**
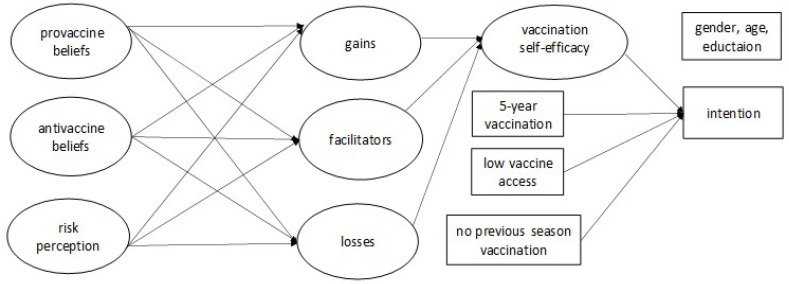
The conceptual model tested in the study.

**Figure 2 vaccines-10-00723-f002:**
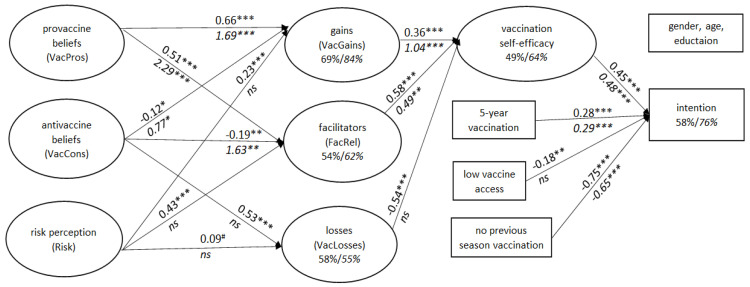
The result of SEM in medics (*n* = 301; path coefficients in italics below the arrows) and non-medics (*n* = 408; path coefficients in regular font above the arrows) analyzed for each group separately (showed in one figure for better visualization). Note: *** = *p* < 0.001; ** = *p* < 0.01; * = *p* < 0.05; # = *p* = 0.05; ns = not significant.

**Figure 3 vaccines-10-00723-f003:**
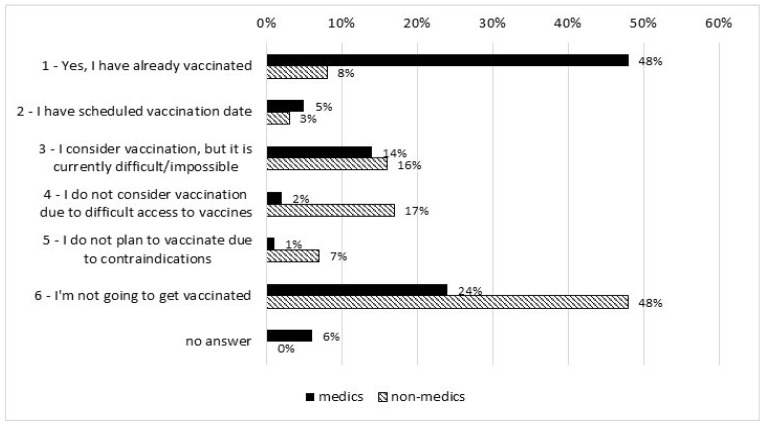
Readiness for influenza vaccination in medics (*n* = 301) and non-medics (*n* = 408).

**Table 1 vaccines-10-00723-t001:** Comparison of predictors in medics (*n* = 301) and non-medics (*n* = 408).

Variable	Group	*M (SD)*	*t*	*df*	*p*
**Pro-vaccine beliefs**	Medics	4.44 (0.85)	15.10	702.86	<0.001
	Non-medics	3.36 (1.06)			
**Antivaccine beliefs**	Medics	1.67 (0.89)	−15.33	681.59	<0.001
	Non-medics	2.76 (1.00)			
**Risk perception**	Medics	2.62 (0.88)	−0.94	707	0.346
	Non-medics	2.68 (0.83)			
**Perceived gains**	Medics	4.16 (0.89)	12.55	706	<0.001
	Non-medics	3.25 (0.93)			
**Perceived losses**	Medics	2.10 (0.82)	−14.24	642.14	<0.001
	Non-medics	2.99 90.81)			
**Facilitators’ relevance**	Medics	3.36 (1.06)	4.30	632.35	<0.001
	Non-medics	3.02 (1.02)			
**Vaccination self-efficacy**	Medics	3.69 (1.48)	10.10	546.64	<0.001
	Non-medics	3.02 (1.02)			

**Table 2 vaccines-10-00723-t002:** Summary of comparison of fit indices of the model between medics and non-medics.

	Chi-Square	RMSEA	Probability RMSEA ≤ 0.05	CFI	TLI	SRMR
Medics	1844.897 ***, df = 1076	0.049	0.708	0.922	0.916	0.063
Non-medics	2162.701 ***, df = 1076	0.050	0.549	0.926	0.920	0.066

Note: *** = *p* < 0.001.

**Table 3 vaccines-10-00723-t003:** Specific indirect effects between predictors in medics and non-medics (only significant).

Effect (From–To)	Group	Estimate	*SE*	*P*
**FacRel–VacSE–Intention/vac**	Medics	0.235	0.076	0.002
	Non-medics	0.260	0.043	<0.001
**VacPros–VacGains–VacSE**	Medics	1.764	0.453	<0.001
	Non-medics	0.239	0.066	<0.001
**VacPros–FacRel–VacSE**	Medics	1.120	0.531	0.035
	Non-medics	0.296	0.048	<0.001
**VacCons–VacGains–VacSE**	Medics	0.798	0.390	0.040
	Non-medics	−0.042	0.022	0.056
**VacCons–VacLosses–VacSE**	Medics	–	–	–
	Non-medics	−0.286	0.085	0.001
**VacCons–FacRel–VacSE**	Medics	–	–	–
	Non-medics	−0.107	0.040	0.007
**Risk–VacLosses–VacSE**	Medics	–	–	–
	Non-medics	0.083	0.030	0.006
**Risk–FacRel–VacSE**	Medics	–	–	–
	Non-medics	0.249	0.052	<0.001

Note: VacPros = pro-vaccination beliefs; VacCons = antivaccination beliefs; Risk = risk perception; VacGains = outcome expectation of perceived gains; VacLosses = outcome expectation of perceived losses; FacRel = facilitators’ relevance; Vac SE = vaccination self efficacy; Intention/vac = intention/vaccination.

## Data Availability

Not applicable.

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
