# Peer review of "Medics as a Positive Deviant in Influenza Vaccination: The Role of Vaccine Beliefs, Self-Efficacy and Contextual Variables"

_vaccines, 2022, doi:10.3390/vaccines10050723_

Round 1

Reviewer 1 Report

I read Włodarczyk's paper entitled, "Physicians as Positive Deviants in Influenza Vaccination: The Role of Vaccine Beliefs, Self-Efficacy, and Contextual Variables."

As much as the paper focuses on a topic of extreme interest, even considering the insufficient data reported by the authors regarding the average influenza vaccination rate for the years 2019/2020 in Poland (4.1%), I must say that both the methodological and stylistic development of the paper in question makes it unacceptable for publication in my opinion.

I am now going to specify better some of the general criticalities, as I cannot go into the details of a very confusing paper that mixes together different parts without having a clear distinction between the things that need to be put into the introduction, methods, results and discussion.

1. Introduction: the introduction is a long dissertation without rhyme or reason in which we first talk about WHO data, then we say why the methodological choices of the paper, then we advance various hypotheses, and finally, without indicating a real aim, we go to define what (I think) are the basic assumptions of the next SEM model. I specify: from line 88 to 104 they describe the conceptual model they hypothesize. I would move this in the introduction to the methods and maybe add figure with only the conceptual model, I do not think it makes sense in the introduction to refer to figure 2 that is a result.

2. The methods and results are apparently very detailed. However I have several things to point out. First of all it is not clear to me how the questionnaire was constructed. The references to which the authors refer are generic. I believe that the authors should have also made the whole questionnaire available to the reader so that one would understand the questions as they were constructed. Regarding the collection I have to say I was shocked that while the physicians filled out a CATI, the general popualtion instead did it sel administered. The sample is not only small (700 people barely) but also extremely inhomogeneous, as noted by the statistical analysis (but not discussed nor even put into limits by the authors). Moreover, how were the physicians recruited? I think the authors absolutely need to clarify these key steps for the reader. The SEM model that the authors use is hardly intelligible. It also seems to add both physicians and non-physicians together. I don't think this is methodologically correct. In the statistical analysis or in the legend of Figure 2 they need to better specify how they reported the results of the SEM model. Finally, there is never any mention of what software they used for the analysis

3. I will not spend many words on the discussion which is very weak even by virtue of the unsound results. Also it is not clear, exactly like the introduction, what the authors' real goal is. Also, in general, you don't end a discussion with limitations and a conclusion paragraph would be in order.

The English of the paper is equally weak, and would need revision by a native speaker. The only thing I can think of to give this paper a chance is for the authors to completely rewrite it from top to bottom following the guidelines of the medical literature (here they find a reference: http://www.icmje.org/recommendations/browse/manuscript-preparation/preparing-for-submission.html) and then make an eventual resubmission.

Author Response

Dear Sir or Madam,

Thank you very much for the opportunity to revise our manuscript. In response to the reviews we trying to use all suggestions to improve its quality and convince that it is worth publishing. Below we present our responses to all Reviewers’ comments. In majority we followed them and included modifications in the text. In single cases we presented our view of the issue trying to explain it. In consequence, the current version of the manuscript contains a number of changes in all parts of it (visible in track mode), especially in the Materials and Methods section and the Discussion section. This led also to changes in references. As suggested by the Reviewers we replaced Figures and added the supplementary file with the questionnaire (translated version). We hope these responses and modifications will be satisfactory.

Below, please find review 1 that was divided into sections. Each section contains our comment. 

We can ensure that before submission, the manuscript has been professionally proofread by Proof-Reading-Service.com, United Kingdom and the company is ready to provide us with a certificate confirming that our article has been proofread by PRS. Regardless of it, if it is necessary we are ready to perform it once again anytime.

Dr. Dorota Włodarczyk

Dr. Urszula Ziętalewicz

Reviewer 2 Report

The manuscript presented has a solid bibliographic basis. The method is well delineated and well described. Data analysis is extensive and represents several aspects of the study. The study is based on two medical and non-medical groups and their perceptions of Influenza vaccination

My considerations are:

  1. Accurately representation of sampling. The sampling – in this paper individuals are targeted to focus or represent the attitudes, interests, or attributes about Influenza vaccination - divided into medics and non-medics participants has a significative bias.
  2. As explained the medics group shows a significant difference in the distribution of education. They had higher education (Master’s degree Bachelor’s degree). Among non-medics group, the majority had secondary education, followed by primary or vocational education and higher education There are also differences of living location;
  3. The results of the study indicate obvious differences: who knows the vaccine and epidemiological principles and who does not know those principles and is subject to the influence of the media, political uses, confusing and contradictory messages, etc.
  4. My suggestion is that only the group of medics should be studied. Why in a group with access to high level of education, academic training in sciences, higher socio-economic-cultural conditions, there is so much discrepancy and inertia in relation to vaccination.

Author Response

(The authors gave the same response as above.)

Reviewer 3 Report

Dear Editor & Authors,

Thank you kindly for asking me to review the manuscript titled “Medics as a positive deviant in influenza vaccination: The role of vaccine beliefs, self-efficacy and contextual variables” by Drs WÅ‚odarczyk and ZiÄ™talewicz from the Medical University of Warsaw (note the word Medical is spelled wrong in the manuscript).  

This article is very pertinent at his time were the discussion about vaccination is in the forefront and examines the vaccination compliance of a perceived “educated about their benefit” special societal group which is us physicians. What this manuscript basically shows to me is that “the king is naked” and that despite the fact that medics performed better than the non-medic group there was still only a 48% rate of vaccination with a significant amount of almost 30% of medics stating that they do not wish to get vaccinated (note to authors – figure 1 is really nice in showing the comparison between the groups but for clarity and making it easier to numerically compare, I would suggest the percentages be added to each bar in the figure! It is a simple function in Excel were I presume the graph was created)

The manuscript is well written and presented with nice clear tables and figures (I would suggest however that, figure 2 be enlarged a bit because it is difficult to see and read). The language and structure is good, understandable with only minor syntax mistakes (note the word medical in the affiliations). The methodology and conduct of the study is also good. I only have some minor issues – suggestions to make to improve the work prior to final recommendation (which I believe should be positive):

1. The modules measuring gains (VacGains) and losses (VacLosses). and the 8 item subscales should be presented in the text or as a list table and not as a supplemental file to make it easier for the reader to evaluate the type of questions used.

2. Was a sample size calculation or power analysis performed prior to study commencement to elucidate if the number of patients queried/recruited is adequate to produce statistically significant results? This is what it appears to be a prospective study so it should have been done at the designing stage!

Thank you again for asking me to review this work. It is quite interesting and I am awaiting a minor revised version to render a positive recommendation.

Kind regards and wishing well to all.

Author Response

(The authors gave the same response as above.)

Round 2

Reviewer 1 Report

I would like to thank the authors for having made some improvements. 

Nevertheless the construction of the sample as well as the analysis and study design are flawed and insufficient for considering it acceptable in the journal, in my opinion.

Author Response

We would like to thank very much for reviewing the resubmitted version of our manuscript. Although the opinion was negative we would like to defend the concept and the value our study. It is very difficult to refer accurately to very general one-sentence comment (and introduce specific changes in the manuscript), but:

  1. All aspects that are subject to criticism (the construction of the sample as well as the analysis and study design) were the consequence of the general concept of the study: to compere those who perform vaccinations noticeable better than general population in searching for specific mechanisms potentially related to better vaccination rate.
  2. The key aim of the study was to compare those mechanisms between the groups of medics and non-medic. These mechanisms (expressed in hypotheses) refer to psychosocial variables which role in vaccination has been previously confirmed, knowing that medical education and practice are not the sufficient, although important condition of vaccination intention.
  3. To establish the proper sample sizes of the groups we performed a priori samples size calculation (using two methods knowing that there are mixed experts opinions in this matter). We met these criteria.
  4. To compare these groups we developed the tool according to the current psychometric recommendations. The tool met the expected psychometric criteria. We have confirmed this in various ways: the Cronbach’s coefficients, confirmatory factor analysis, invariance analysis.
  5. To our knowledge (confirmed by professional survey company) we adopted the only possible way of data collection and its methodological correctness was confirmed by invariance analysis – the data were eligible for comparison.
  6. We used advanced statistics allowing to verify the hypotheses regarding direct and indirect effects assumed in the theoretical model.
  7. The results showed differences in the mechanisms related to (and potentially responsible for) vaccination intention between groups:

- delivering new knowledge on how to modify messages and action aimed at improving vaccination rate in general public

- delivering unique knowledge on medics considered in this study as the positive deviant in vaccination implementation.

  1. Although the study was exploratory, its outcomes are unique in the above aspects and that is why we believe they are worth publishing just in this Journal and disseminate among specialists in the field.

Reviewer 2 Report

Dear authors

As presented in the previous review the manuscript is of quality, however, it does not present new information of relevance or specific propositions that contribute to greater population awareness of vaccination adeb. I insist that comparing groups with expertise and without expertise in a given area produces obvious results. It is not surprising that the group with expertise (medics) present, in part, common responses with the general population. We saw that during the COVID-19 pandemic the isolated medical opinions were inappropriate, wrong, and generated controversial information that harmed vaccination campaigns, prevention care and others.

My previous suggestion remains: to study the group with expertise to understand how the medical community deals with epidemiological data, with the concepts of immunity and with responsibility for the dissemination of information to the population.

Author Response

We would like to thank very much for reviewing the resubmitted version of the manuscript. We agree with the final recommendation to study only medics (the group with expertise) to understand how the medical community deals with epidemiological data, with the concepts of immunity and with responsibility for the dissemination of information to the population and we intend to do it using other data. However, we are not able to use this recommendation to improve this manuscript. As the main criticism refers to the general idea of the study (not to its quality), we would like to defend this idea:

  1. We would like to underline that our study focused not on how to increase the vaccination awareness in the population, but on understanding the mechanisms related to vaccination intention. Only knowing these mechanisms makes it possible to develop specific propositions of how to promote vaccination matched to the reality.
  2. Considering medics as a positive deviant is a chance to find out what is an advantage for those who get vaccinated better, aside from education which, as we know, is not a sufficient factor (this is maybe disappointing, but observed recently).
  3. Thanks to understanding mechanisms related to better vaccination performance in medics, we get to know how to transfer medical academic knowledge into health literacy in general population – we are not able to educate the whole general population medically (at the academic level), but we may discover what elements (apart from knowledge) and mechanisms are related to stronger vaccination intention – and this is why it makes sense to compare these seemingly incomparable groups.
  4. We don’t agree that our study produced obvious results. The fact that the groups did differ in the levels of the majority of psychosocial variables (apart from risk perception) in not surprising but it was not the essence of the study (this are only descriptive characteristics). We focused on mechanisms related to vaccination intention and we confirmed the differences between groups and they were not obvious, for example:

- the groups didn’t differ in the level of risk perception but its role in vaccination intention was completely different in the study groups (significant only in non-medics)

- as for antivaccination beliefs, their effects in groups were of the opposite directions and their positive effects in medics (despite lower level than in non-medics) are very surprising and provoking for further studies

- the level of facilitators relevance was higher in medics but its role in prediction was stronger in non-medics

- the study confirmed the negative effect of vaccination losses – this result seems quite obvious in numbers, but what if it is omitted or insufficiently taken into account in information campaigns?

  1. We admit that the study was exploratory, but to our best knowledge is the only one using positive deviant approach in vaccination – the results of the comparisons presented in the manuscript can be valuable inspiration how to learn from those who perform better.

Round 3

Reviewer 2 Report

The author's response was convincing. Paper approved.